# Enhanced First and Zeroth Order Variance Reduced Algorithms for Min-Max Optimization

## Abstract

Min-max optimization captures many important machine learning problems such as robust adversarial learning and inverse reinforcement learning, and nonconvex-strongly-concave min-max optimization has been an active line of research. Specifically, a novel variance reduction algorithm SREDA was proposed recently by (Luo et al. 2020) to solve such a problem, and was shown to achieve the optimal complexity dependence on the required accuracy level $\epsilon$. Despite the superior theoretical performance, the convergence guarantee of SREDA requires stringent initialization accuracy and an $\epsilon$-dependent stepsize for controlling the per-iteration progress, so that SREDA can run very slowly in practice. This paper develops a novel analytical framework that guarantees the SREDA's optimal complexity performance for a much enhanced algorithm SREDA-Boost, which has less restrictive initialization requirement and an accuracy-independent (and much bigger) stepsize. Hence, SREDA-Boost runs substantially faster in experiments than SREDA. We further apply SREDA-Boost to propose a zeroth-order variance reduction algorithm named ZO-SREDA-Boost for the scenario that has access only to the information about function values not gradients, and show that ZO-SREDA-Boost outperforms the best known complexity dependence on $\epsilon$. This is the first study that applies the variance reduction technique to zeroth-order algorithm for min-max optimization problems.

## 1 Introduction

Min-max optimization has attracted significant growth of attention in machine learning as it captures several important machine learning models and problems including generative adversarial networks (GANs) Goodfellow et al. (2014), robust adversarial machine learning Madry et al. (2018), imitation learning Ho & Ermon (2016), etc. Min-max optimization typically takes the following form

$$\min_{x \in \mathbb{R}^{d_1}} \max_{y \in \mathbb{R}^{d_2}} f(x,y), \quad \text{where} \quad f(x,y) \triangleq \begin{cases} \mathbb{E}[F(x,y;\xi)] & \text{(online case)} \\ \frac{1}{n}\sum_{i=1}^{n} F(x,y;\xi_i) & \text{(finite-sum case)} \end{cases} \tag{1}$$

where $f(x,y)$ takes the expectation form if data samples $\xi$ are taken in an online fashion, and $f(x,y)$ takes the finite-sum form if a dataset of training samples $\xi_i$ for $i = 1, \ldots, n$ are given in advance.

This paper focuses on the nonconvex-strongly-concave min-max problem, in which $f(x,y)$ is nonconvex with respect to $x$ for all $y \in \mathbb{R}^{d_2}$, and $f(x,y)$ is $\mu$-strongly concave with respect to $y$ for all $x \in \mathbb{R}^{d_1}$. The problem then takes the following equivalent form:

$$\min_{x \in \mathbb{R}^{d_1}} \left\{ \Phi(x) \triangleq \max_{y \in \mathbb{R}^{d_2}} f(x,y) \right\}. \tag{2}$$

The objective function $\Phi(\cdot)$ in eq. (2) is nonconvex in general, and hence algorithms for solving eq. (2) are expected to attain an approximate (i.e., $\epsilon$-accurate) first-order stationary point. The convergence of deterministic algorithms for solving eq. (2) has been established in Jin et al. (2019); Nouiehed et al. (2019); Thekumparampil et al. (2019); Lu et al. (2020). SGD-type of stochastic algorithms have also been proposed to solve such a problem more efficiently, including SGDmax Jin et al. (2019),

PGSMD Rafique et al. (2018), and SGDA Lin et al. (2019), which respectively achieve the overall complexity of $\mathcal{O}(\kappa^3\epsilon^{-4}\log(1/\epsilon))^1$, $\mathcal{O}(\kappa^3\epsilon^{-4})$, and $\mathcal{O}(\kappa^3\epsilon^{-4})$.

Furthermore, several variance reduction methods have been proposed for solving eq. (2) for the nonconvex-strongly-concave case. PGSVRG Rafique et al. (2018) adopts a proximally guided SVRG method and achieves the overall complexity of $\mathcal{O}(\kappa^3\epsilon^{-4})$ for the online case and $\mathcal{O}(\kappa^2 n\epsilon^{-2})$ for the finite-sum case. Wai et al. (2019) converted the value function evaluation problem to a specific min-max problem and applied SAGA to achieve the overall complexity of $\mathcal{O}(\kappa n\epsilon^{-2})$ for the finite-sum case. More recently, Luo et al. (2020) proposed a novel nested-loop algorithm named Stochastic Recursive Gradient Descent Ascent (SREDA), which adopts SARAH/SPIDER-type Nguyen et al. (2017a); Fang et al. (2018) of recursive variance reduction method (originally designed for solving the minimization problem) for designing gradient estimators to update both $x$ and $y$. Specifically, $x$ takes the normalized gradient update in the outer-loop and each update of $x$ is followed by an entire inner-loop updates of $y$. Luo et al. (2020) showed that SREDA achieves an overall complexity of $\mathcal{O}(\kappa^3\epsilon^{-3})$ for the online case in eq. (1), which attains the optimal dependence on $\epsilon$ Arjevani et al. (2019). For the finite-sum case, SREDA achieves the complexity of $\mathcal{O}(\kappa^2\sqrt{n}\epsilon^{-2} + n + (n+k)\log(\kappa/\epsilon))$ for $n \geq \kappa^2$, and $\mathcal{O}((\kappa^2 + \kappa n)\epsilon^{-2})$ for $n \leq \kappa^2$.

Although SREDA achieves the optimal complexity performance in theory, two issues may substantially degrade its practice performance. (1) SREDA has a stringent requirement on the initialization accuracy $\zeta = \kappa^{-2}\epsilon^2$, which hence requires $\mathcal{O}(\kappa^2\epsilon^{-2}\log(\kappa/\epsilon))$ gradient estimations in the initialization, and is rather costly in the high accuracy regime (i.e., for small $\epsilon$). (2) The convergence of SREDA requires the stepsize to be substantially small, i.e., at the $\epsilon$-level with $\alpha_t = \mathcal{O}(\min\{\epsilon/(\kappa\ell\|v_t\|_2), 1/(\kappa\ell)\})$, which restricts each iteration to make only $\epsilon$-level progress with $\|x_{t+1} - x_t\|_2 = \mathcal{O}(\epsilon/\kappa\ell)$. Consequently, SREDA can run very slowly in practice.

- Thus, a vital question arising here is whether we can guarantee the same optimal complexity performance of SREDA even if we considerably relax its initialization (i.e., much bigger than $\mathcal{O}(\epsilon^2)$) and enlarge its stepsize (i.e., much bigger than $\mathcal{O}(\epsilon)$). The answer is highly nontrivial because the original analysis framework for SREDA in Luo et al. (2020) critically relies on these restrictions. The first focus of this paper is on developing a novel analytical framework to guarantee that such enhanced SREDA continues to hold the SREDA's optimal complexity performance.

Furthermore, in many machine learning scenarios, min-max optimization problems need to be solved without the access of the gradient information, but only the function values, e.g., in multi-agent reinforcement learning with bandit feedback Wei et al. (2017); Zhang et al. (2019) and robotics Wang & Jegelka (2017); Bogunovic et al. (2018). This motivates the design of zeroth-order (i.e., gradient-free) algorithms. For nonconvex-strongly-concave min-max optimization, Liu et al. (2019) studied a constrained problem and proposed ZO-min-max algorithm that achieves the computational complexity of $\mathcal{O}((d_1 + d_2)\epsilon^{-6})$. Wang et al. (2020) designed ZO-SGDA and ZO-SGDMSA, where ZO-SGDMA achieves the best known query complexity of $\mathcal{O}((d_1 + d_2)\kappa^2\epsilon^{-4}\log(1/\epsilon))$ among the zeroth-order algorithms for this problem. All of the above studies are of SGD-type, and no efforts have been made on developing variance reduction zeroth-order algorithms for nonconvex-strongly-concave min-max optimization to further improve the query complexity.

- The second focus of this paper is on applying the aforementioned enhanced SREDA algorithm to design a *zeroth-order variance reduced algorithm* for nonconvex-strongly-concave min-max problems, and further characterizing its complexity guarantee which we anticipate to be orderwisely better than that of the existing stochastic algorithms.

## 1.1 MAIN CONTRIBUTIONS

This paper first studies an enhanced SREDA algorithm, called SREDA-Boost, which improves SREDA from two aspects. (1) For the initialization, SREDA-Boost requires only an accuracy of $\zeta = \kappa^{-1}$, which is much less stringent than that of $\zeta = \kappa^{-2}\epsilon^2$ required by SREDA. (2) SREDA-Boost allows an accuracy-independent stepsize $\alpha = \mathcal{O}(1/(\kappa\ell))$, which is much larger than the $\epsilon$-level stepsize $\alpha = \mathcal{O}(\min\{\epsilon/(\kappa\ell\|v_t\|_2), 1/(\kappa\ell)\})$ adopted by SREDA. Hence, SREDA-Boost can run much faster than SREDA.

---

[1]The constant $\kappa = \ell/\mu$, where $\mu$ is the strong concavity parameter of $f(x, \cdot)$, and $\ell$ is the Lipschitz constant of the gradient of $f(x, y)$ as defined in Assumption 2. Typically, $\kappa$ is much larger than one.

Table 1: Comparison of stochastic algorithms for nonconvex-strongly-concave min-max problems

| Type[‡] | Algorithm | Stepsize[♯] | Initialization Complexity | Complexity[◇, ♡] |
|---|---|---|---|---|
| FO | SGDmax | $\Theta(\kappa^{-1}\ell^{-1})$ | N/A | $\mathcal{O}(\kappa^3\epsilon^{-4}\log(\frac{1}{\epsilon}))$ |
| | SGDA | $\Theta(\kappa^{-2}\ell^{-1})$ | N/A | $\mathcal{O}(\kappa^3\epsilon^{-4})$ |
| | PGSMD | $\Theta(\kappa^{-2})$ | N/A | $\mathcal{O}(\kappa^3\epsilon^{-4})$ |
| | PGSVRG | $\Theta(\kappa^{-2})$ | N/A | $\mathcal{O}(\kappa^3\epsilon^{-4})$ |
| | SREDA | $\Theta(\min\{\frac{\epsilon}{\kappa\ell\|v_t\|_2}, \frac{1}{\kappa\ell}\})$ | $\mathcal{O}(\kappa^2\epsilon^{-2}\log(\frac{\kappa}{\epsilon}))$ | $\mathcal{O}(\kappa^3\epsilon^{-3})$ |
| | SREDA-Boost | $\Theta((\kappa^{-1}\ell^{-1}))$ | $\mathcal{O}(\kappa\log(\kappa))$ | $\mathcal{O}(\kappa^3\epsilon^{-3})^{\dagger}$ |
| ZO | ZO-min-max | $\Theta(\kappa^{-1}\ell^{-1})$ | N/A | $\mathcal{O}((d\epsilon^{-6})^{\natural}$ |
| | ZO-SGDA | $\Theta(\kappa^{-4}\ell^{-1})$ | N/A | $\mathcal{O}(d\kappa^5\epsilon^{-4})$ |
| | ZO-SGDMSA | $\Theta(\kappa^{-1}\ell^{-1})$ | N/A | $\mathcal{O}(d\kappa^2\epsilon^{-4}\log(\frac{1}{\epsilon}))$ |
| | ZO-SREDA-Boost | $\Theta(\kappa^{-1}\ell^{-1})$ | $\mathcal{O}(\kappa\log(\kappa))$ | $\mathcal{O}(d\kappa^3\epsilon^{-3})$ |

[†] We clarify that SREDA-Boost should not be expected to improve the complexity order of SREDA, because SREDA already achieves the optimal complexity. Rather, SREDA-Boost improves upon SREDA by much more relaxed requirements on initialization and stepsize to achieve such optimal performance.

[‡] "FO" stands for "First-Order", and "ZO" stands for "Zeroth-Order".

[♯] We include only the stepsize for updating $x_t$ for comparison.

[◇] The complexity for first-order algorithms refer to the total gradient computations to attain an $\epsilon$-stationary point, and for zeroth-order algorithms refers to the total function value queries.

[♡] We include only the complexity in the online case in the table, because many studies did not cover the finite-sum case. We comment on the finite-sum case in Section 4 and Section 5.2.

[♮] We define $d = d_1 + d_2$.

The **first contribution** of this paper lies in developing a new analysis technique to provide the computational complexity guarantee for SREDA-Boost, establishing that even with considerably relaxed conditions on the initialization and stepsize, SREDA-Boost achieves the same optimal complexity performance as SREDA. The analysis technique of SREDA in Luo et al. (2020) does not handle such a case, because the proof highly relies on the stringent initialization and stepsize requirements. Central to our new analysis framework is the novel approach for bounding two interconnected stochastic error processes: tracking error and gradient estimation error (see Section 4 for their formal definitions), which take three steps: bounding the two error processes accumulatively over the entire algorithm execution, decoupling these two inter-related stochastic error processes, and establishing each of their relationships with the accumulative gradient estimators.

The **second contribution** of this paper lies in proposing the zeroth-order variance reduced algorithm ZO-SREDA-Boost for nonconvex-strongly-conconve min-max optimization when the gradient information is not accessible. For the online case, we show that ZO-SREDA-Boost achieves an overall query complexity of $\mathcal{O}((d_1 + d_2)\kappa^3\epsilon^{-3})$, which outperforms the best known complexity (achieved by ZO-SGDMSA Wang et al. (2020)) in the case with $\epsilon \leq \kappa^{-1}$. For the finite-sum case, we show that ZO-SREDA-Boost achieves an overall query complexity of $\mathcal{O}((d_1+d_2)(\kappa^2\sqrt{n}\epsilon^{-2}+n) + d_2(\kappa^2+\kappa n)\log(\kappa))$ when $n \geq \kappa^2$, and $\mathcal{O}((d_1+d_2)(\kappa^2+\kappa n)\kappa\epsilon^{-2})$ when $n \leq \kappa^2$. This is the first study that applies the variance reduction method to zeroth-order nonconvex-stronlgy-concave min-max optimization.

## 1.2 RELATED WORK

Due to the vast amount of studies on min-max optimization and the variance reduced algorithms, we include below only the studies that are highly relevant to this work.

Variance reduction methods for min-max optimization are highly inspired by those for conventional minimization problems, including SAGA Defazio et al. (2014); Reddi et al. (2016), SVRG Johnson & Zhang (2013); Allen-Zhu & Hazan (2016); Allen-Zhu (2017), SARAH Nguyen et al. (2017a;b; 2018), SPIDER Fang et al. (2018), SpiderBoost Wang et al. (2019), etc. But the convergence analysis for min-max optimization is much more challenging, and is typically quite different from their counterparts in minimization problems.

For *strongly-convex-strongly-concave min-max optimization*, Palaniappan & Bach (2016) applied SVRG and SAGA to the finite-sum case and established a linear convergence rate, and Chavdarova et al. (2019) proposed SVRE later to obtain a better bound. When the condition number of the problem is very large, Luo et al. (2019) proposed a proximal point iteration algorithm to improve the

performance of SAGA. For some special cases, Du et al. (2017); Du & Hu (2019) showed that the linear convergence rate of SVRG can be maintained without the strongly-convex or strongly concave assumption. Yang et al. (2020) applied SVRG to study the min-max optimization under the two-sided Polyak-Lojasiewicz condition.

*Nonconvex-strongly-concave min-max optimization* is the focus of this paper. As we discuss at the beginning of the introduction, the SGD-type algorithms have been developed and studied, including SGDmax Jin et al. (2019), PGSMD Rafique et al. (2018), and SGDA Lin et al. (2019). Several variance reduction methods have also been proposed to further improve the performance, including PGSVRG Rafique et al. (2018), the SAGA-type algorithm for min-max optimization Wai et al. (2019), and SREDA Luo et al. (2020). Particularly, SREDA has been shown in Luo et al. (2020) to achieve the optimal complexity dependence on $\epsilon$. This paper further provides the convergence guarantee for SREDA-Boost (which enhances SREDA with relaxed initialization and much larger stepsize) by developing a new analysis technique.

While SGD-type zeroth-order algorithms have been studied for min-max optimization, such as Menickelly & Wild (2020); Roy et al. (2019) for convex-concave min-max problems and Liu et al. (2019); Wang et al. (2020) for nonconvex-strongly-concave min-max problems, variance reduced algorithms have not been developed for *zeroth-order min-max optimization* so far. This paper proposes the first such an algorithm named ZO-SREDA-Boost for nonconvex-strongly-concave min-max optimization, and established its complexity performance that outperforms the existing comparable algorithms (see Table 1).

## 2 NOTATION AND PRELIMINARIES

In this paper, we use $\|\cdot\|_2$ to denote the Euclidean norm of vectors. For a finite set $\mathcal{S}$, we denote its cardinality as $|\mathcal{S}|$. For a positive integer $n$, we denote $[n] = \{1, \cdots, n\}$. We assume that the min-max problem eq. (2) satisfies the following assumptions, which have also been adopted by Luo et al. (2020) for SREDA. We slightly abuse the notation $\xi$ below to represent the random index in both the online and finite-sum cases, where in the finite-sum case, $\mathbb{E}_\xi[\cdot]$ is with respect to the uniform distribution over $\{\xi_1, \cdots, \xi_n\}$.

**Assumption 1.** *The function $\Phi(\cdot)$ is lower bounded, i.e., we have $\Phi^* = \inf_{x \in \mathbb{R}^{d_1}} \Phi(x) > -\infty$.*

**Assumption 2.** *The component function $F$ has an averaged $\ell$-Lipschitz gradient, i.e., for all $(x, y)$, $(x', y') \in \mathbb{R}^{d_1} \times \mathbb{R}^{d_2}$, we have $\mathbb{E}_\xi[\|\nabla F(x, y; \xi) - \nabla F(x', y'; \xi)\|_2^2] \leq \ell^2(\|x - x'\|_2^2 + \|y - y'\|_2^2)$.*

**Assumption 3.** *The function $f$ is $\mu$-strongly-concave in $y$ for any $x \in \mathbb{R}^{d_1}$, and the component function $F$ is concave in $y$, i.e., for any $x \in \mathbb{R}^{d_1}$, $y, y' \in \mathbb{R}^{d_2}$ and $\xi$, we have $f(x, y) \leq f(x, y') + \langle \nabla_y f(x, y'), y - y' \rangle - \frac{\mu}{2}\|y - y'\|_2$, and $F(x, y; \xi) \leq F(x, y'; \xi) + \langle \nabla_y F(x, y'; \xi), y - y' \rangle$.*

**Assumption 4.** *The gradient of each component function $F(x, y; \xi)$ has a bounded variance, i.e., there exists a constant $\sigma > 0$ such that for any $(x, y) \in \mathbb{R}^{d_1 \times d_2}$, we have $\mathbb{E}_\xi[\|\nabla F(x, y; \xi) - \nabla f(x, y)\|_2^2] \leq \sigma^2 < \infty$.*

Since $\Phi$ is nonconvex in general, it is NP-hard to find its global minimum. The goal here is to develop stochastic gradient algorithms that output an $\epsilon$-stationary point as defined below.

**Definition 1.** *The point $\bar{x}$ is called an $\epsilon$-stationary point of the differentiable function $\Phi$ if $\|\nabla \Phi(\bar{x})\|_2 \leq \epsilon$, where $\epsilon$ is a positive constant.*

## 3 SREDA AND SREDA-BOOST ALGORITHMS

We first introduce the SREDA algorithm proposed in Luo et al. (2020), and then describe an enhanced algorithm SREDA-Boost that we study in this paper.

SREDA (see **Option I** in Algorithm 1) utilizes the variance reduction techniques proposed in SARAH Nguyen et al. (2017a) and SPIDER Fang et al. (2018) for minimization problems to construct the gradient estimator recursively for min-max optimization. Specifically, the parameters $x_t$ and $y_t$ are updated in a nested loop fashion: each update of $x_t$ in the outer-loop is followed by $(m + 1)$ updates of $y_t$ over one entire inner loop. Furthermore, the outer-loop updates of $x_t$ is divided into epochs for variance reduction. Consider a certain outer-loop epoch $t = \{(n_t - 1)q, \cdots, n_t q - 1\}$ ($1 \leq n_t < \lceil T/q \rceil$ is a positive integer). At the beginning of such an epoch, the gradients are evaluated

with a large batch size $S_1$ (see line 6 in Algorithm 1). Then, for each subsequent outer-loop iteration, an inner loop of ConcaveMaximizer (see Algorithm 2) recursively updates the gradient estimators for $\nabla_x f(x, y)$ and $\nabla_y f(x, y)$ with a small batch size $S_2$. Note that although the inner loop does not update $x$, the gradient estimator $\nabla_x f(x, y)$ is updated in the inner loop. With such a variance reduction technique, SREDA outperforms all previous algorithms for nonconvex-strongly-concave min-max problems (see Table 1), and was shown to achieve the optimal dependency on $\epsilon$ in complexity Luo et al. (2020).

---

**Algorithm 1** SREDA and SREDA-Boost

1: **Input:** $x_0$, initial accuracy $\zeta$, learning rate $\alpha_t$, $\beta = \mathcal{O}(\frac{1}{\ell})$, batch size $\mathcal{S}_1$, $\mathcal{S}_2$ and periods $q, m$.
2: **Option I (SREDA):** $\zeta = \kappa^{-2}\epsilon^2$; **Option II (SREDA-Boost):** $\zeta = \kappa^{-1}$
3:      **Initialization:** $y_0 = \text{iSARAH}(-f(x_0, \cdot), \zeta)$ (see Appendix B.2 for iSARAH($\cdot$))
4: **for** $t = 0, 1, ..., T - 1$ **do**
5:   **if** $\text{mod}(t, q) = 0$ **then**    draw $S_1$ samples $\{\xi_1, \cdots, \xi_{S_1}\}$
6:     $v_t = \frac{1}{S_1}\sum_{i=1}^{S_1} \nabla_x F(x_t, y_t, \xi_i), \quad u_t = \frac{1}{S_1}\sum_{i=1}^{S_1} \nabla_y F(x_t, y_t, \xi_i)$
7:   **else**
8:     $v_t = \tilde{v}_{t-1, \tilde{m}_{t-1}}, \quad u_t = \tilde{u}_{t-1, \tilde{m}_{t-1}}$
9:   **end if**
10:   **Option I (SREDA):** $\alpha_t = \min\{\frac{\epsilon}{\ell\|v_t\|_2}, \frac{1}{2\ell}\}\mathcal{O}(\frac{1}{\kappa})$; **Option II (SREDA-Boost):** $\alpha_t = \alpha = \mathcal{O}(\frac{1}{\kappa\ell})$
11:     $x_{t+1} = x_t - \alpha_t v_t$
12:     $y_{t+1} = \text{ConcaveMaximizer}(t, m, \mathcal{S}_2)$
13: **end for**
14: **Output:** $\hat{x}$ chosen uniformly at random from $\{x_t\}_{t=0}^{T-1}$

---

**Algorithm 2** ConcaveMaximizer$(t, m, \mathcal{S}_2)$

1: **Initialization:** $\tilde{x}_{t,-1} = x_t, \tilde{y}_{t,-1} = y_t, \tilde{x}_{t,0} = x_{t+1}, \tilde{y}_{t,0} = y_t, \tilde{v}_{t,-1} = v_t, \tilde{u}_{t,-1} = u_t$
2: Draw $S_2$ samples $\{\xi_1, \cdots, \xi_{S_2}\}$
3: $\tilde{v}_{t,0} = \tilde{v}_{t,-1} + \frac{1}{S_2}\sum_{i=1}^{S_2} \nabla_x F(\tilde{x}_{t,0}, \tilde{y}_{t,0}, \xi_i) - \frac{1}{S_2}\sum_{i=1}^{S_2} \nabla_x F(\tilde{x}_{t,-1}, \tilde{y}_{t,-1}, \xi_i)$
4: $\tilde{u}_{t,0} = \tilde{u}_{t,-1} + \frac{1}{S_2}\sum_{i=1}^{S_2} \nabla_y F(\tilde{x}_{t,0}, \tilde{y}_{t,0}, \xi_i) - \frac{1}{S_2}\sum_{i=1}^{S_2} \nabla_y F(\tilde{x}_{t,-1}, \tilde{y}_{t,-1}, \xi_i)$
5: $\tilde{x}_{t,1} = \tilde{x}_{t,0}, \quad \tilde{y}_{t,1} = \tilde{y}_{t,0} + \beta\tilde{u}_{t,0}$
6: **for** $k = 1, 2, ..., m - 1$ **do**
7:   draw $S_2$ samples $\{\xi_1, \cdots, \xi_{S_2}\}$
8:   $\tilde{v}_{t,k} = \tilde{v}_{t,k-1} + \frac{1}{S_2}\sum_{i=1}^{S_2} \nabla_x F(\tilde{x}_{t,k}, \tilde{y}_{t,k}, \xi_i) - \frac{1}{S_2}\sum_{i=1}^{S_2} \nabla_x F(\tilde{x}_{t,k-1}, \tilde{y}_{t,k-1}, \xi_i)$
9:   $\tilde{u}_{t,k} = \tilde{u}_{t,k-1} + \frac{1}{S_2}\sum_{i=1}^{S_2} \nabla_y F(\tilde{x}_{t,k}, \tilde{y}_{t,k}, \xi_i) - \frac{1}{S_2}\sum_{i=1}^{S_2} \nabla_y F(\tilde{x}_{t,k-1}, \tilde{y}_{t,k-1}, \xi_i)$
10:   $\tilde{x}_{t,k+1} = \tilde{x}_{t,k}, \quad \tilde{y}_{t,k+1} = \tilde{y}_{t,k} + \beta\tilde{u}_{t,k}$
11: **end for**
12: **Output:** $y_{t+1} = \tilde{y}_{t,\tilde{m}_t}$ with $\tilde{m}_t$ chosen uniformly at random from $\{0, 1, \cdots, m\}$

---

Although SREDA achieves the optimal complexity performance in theory, two issues can substantially slow down its practical performance. **(a)** Its initialization $y_0$ needs to satisfy a stringent $\epsilon^2$-level requirement on the accuracy $\mathbb{E}[\|\nabla_y f(x_0, y_0)\|_2^2] \leq \kappa^{-2}\epsilon^2$ (see line 2 in Algorithm 1), which requires as large as $\mathcal{O}(\kappa^2\epsilon^{-2}\log(\kappa/\epsilon))$ stochastic gradient computations Luo et al. (2020). This is quite costly. **(b)** SREDA uses an $\epsilon$-dependent stepsize and applies normalized gradient descent, so that each outer-loop update makes only $\epsilon$-level progress given by $\|x_{t+1} - x_t\|_2 = \mathcal{O}(\epsilon/(\kappa\ell))$. This substantially slows down SREDA. By following the analysis of SREDA, it appears that such choices for initialization and stepsize are necessary to obtain the guaranteed convergence rate.

In this paper, we study SREDA-Boost (see **Option II** in Algorithm 1) that enhances SREDA over the above two issues. **(a)** SREDA-Boost relaxes the initialization requirement to be $\mathbb{E}[\|\nabla_y f(x_0, y_0)\|_2^2] \leq \kappa^{-1}$, which requires only $\mathcal{O}(\kappa\log\kappa)$ gradient computations. This improves the computational cost upon SREDA by a factor of $\tilde{\mathcal{O}}(\kappa\epsilon^{-2})$. **(b)** SREDA-Boost adopts an $\epsilon$-**in**dependent stepsize $\alpha_t = \alpha = \mathcal{O}(1/(\kappa\ell))$ for $x_t$ so that each outer-loop update can make much bigger progress than SREDA. As our experiments in Section 6 demonstrate, SREDA-Boost runs much faster than SREDA.

To provide the convergence guarantee for SREDA-Boost, the analysis of SREDA in Luo et al. (2020) does not apply, because the proof highly depends on the stringent requirements on the

initialization and the stepsize. Thus, this paper provides a new analysis technique for establishing the complexity performance guarantee for SREDA-Boost and further applies it to the gradient-free min-max problems.

## 4 CONVERGENCE ANALYSIS OF SREDA-BOOST

The following theorem provides the computational complexity of SREDA-Boost for finding a first-order stationary point of $\Phi(\cdot)$ with $\epsilon$ accuracy.

**Theorem 1.** *Apply SREDA-Boost to solve the online case of the problem eq. (1). Suppose Assumptions 1-4 hold. Let $\zeta = \kappa^{-1}$, $\alpha = \mathcal{O}(\kappa^{-1}\ell^{-1})$, $\beta = \mathcal{O}(\ell^{-1})$, $q = \mathcal{O}(\epsilon^{-1})$, $m = \mathcal{O}(\kappa)$, $S_1 = \mathcal{O}(\sigma^2\kappa^2\epsilon^{-2})$ and $S_2 = \mathcal{O}(\kappa\epsilon^{-1})$. Then for $T$ to be at least at the order of $\mathcal{O}(\kappa\epsilon^{-2})$, Algorithm 1 outputs $\hat{x}$ that satisfies $\mathbb{E}[\|\nabla\Phi(\hat{x})\|_2] \leq \epsilon$ with stochastic gradient complexity $\mathcal{O}(\kappa^3\epsilon^{-3})$.*

Furthermore, SREDA-Boost is also applicable to the finite-sum case of the problem eq. (1) by replacing the large batch $S_1$ of samples used in line 6 of Algorithm 1 with the full set of samples.

**Corollary 1.** *Apply SREDA-Boost described above to solve the finite-sum case of the problem eq. (1). Suppose Assumption 1-4 hold. Under appropriate parameter settings given in Appendix B.4, the overall gradient complexity to attain an $\epsilon$-stationary point is $\mathcal{O}(\kappa^2\sqrt{n}\epsilon^{-2} + n + (n + \kappa)\log(\kappa))$ for $n \geq \kappa^2$, and $\mathcal{O}((\kappa^2 + \kappa n)\epsilon^{-2})$ for $n \leq \kappa^2$.*

To compare with SREDA, as shown in Luo et al. (2020), SREDA requires stringent initialization and stepsize selection to achieve the optimal complexity performance. In constrast, Theorem 1 and Corollary 1 show that those strict requirements are not necessary for achieving the optimal performance, and establish that SREDA-Boost achieves the same optimal complexity as SREDA under more relaxed initialization and a much bigger and accuracy-independent stepsize $\alpha$.

The convergence analysis of SREDA-Boost in Theorem 1 is very different from the proof of SREDA in Luo et al. (2020). At a high level, such analysis mainly focuses on bounding two inter-related errors: **tracking error** $\delta_t = \mathbb{E}[\|\nabla_y f(x_t, y_t)\|_2^2]$ that captures how well the output $y_t$ of the inner loop approximates the optimal point $y^*(x_t)$ for a given $x_t$, and **gradient estimation error** $\Delta_t = \mathbb{E}[\|v_t - \nabla_x f(x_t, y_t)\|_2^2 + \|u_t - \nabla_y f(x_t, y_t)\|_2^2]$ that captures how well the stochastic gradient estimators approximate the true gradients. In the analysis of SREDA in Luo et al. (2020), the stringent requirements for initialization and stepsize and the $\epsilon$-level normalized gradient descent update substantially help to bound both errors $\delta_t$ and $\Delta_t$ separately at the $\epsilon$ level for each iteration so that the convergence bound follows. In contrast, this is not applicable to SREDA-Boost which has relaxed and accuracy-**in**dependent initialization and stepsize. Hence, we develop a novel analysis framework to bound the accumulative errors $\sum_{t=0}^{T-1} \delta_t$ and $\sum_{t=0}^{T-1} \Delta_t$ over the entire algorithm execution, and then decouple these two inter-related stochastic error processes and establish their relationships with the accumulative gradient estimators $\sum_{i=0}^{T-1} \mathbb{E}[\|v_t\|_2^2]$. The following proof sketch of Theorem 1 further illustrates our ideas.

The analysis of SREDA-Boost for *min-max* problems is inspired by that for SpiderBoost in Wang et al. (2019) for *minimization* problems, but the analysis here is much more challenging due to the complicated mathematical nature of min-max optimization. Specifically, SpiderBoost needs to handle only one type of the gradient estimation error, whereas SREDA-Boost requires to handle two strongly coupled errors in min-max problems. Hence, the novelty for analyzing SREDA-Boost mainly lies in bounding and decoupling the two errors in order to characterize their impact on the convergence bound.

## 5 ZO-SREDA-BOOST AND CONVERGENCE ANALYSIS

In this section, we study the min-max problem when the gradient information is not available, but only function values can be used for designing algorithms. Based on the first-order SREDA-Boost algorithm, we first propose the zeroth-order variance reduced algorithm called ZO-SREDA-Boost and then provide the convergence analysis for such an algorithm.

## 5.1 ZO-SREDA-BOOST ALGORITHM

The ZO-SREDA-Boost algorithm (see Algorithm 4 in Appendix C.1) shares the same update scheme as SREDA-Boost, but makes the following changes.

(1) In line 3 of SREDA-Boost, instead of using iSARAH, ZO-SREDA-Boost utilizes a zeroth-order algorithm ZO-iSARAH (Algorithm 6 in Appendix C.4) to search an initialization $y_0$.

(2) At the beginning of each epoch in the outer loop (line 6 of SREDA-Boost), ZO-SREDA-Boost utilizes coordinate-wise gradient estimators with a large batch $S_1$ given by $v_t = (1/S_1) \sum_{i=1}^{S_1} \sum_{j=1}^{d_1} (F(x_t + \delta e_j, y_t, \xi_i) - F(x_t - \delta e_j, y_t, \xi_i)) e_j/(2\delta)$ and $u_t = (1/S_1) \sum_{i=1}^{S_1} \sum_{j=1}^{d_2} (F(x_t, y_t + \delta e_j, \xi_i) - F(x_t, y_t - \delta e_j, \xi_i)) e_j/(2\delta)$, where $e_j$ denotes the $j$-th canonical unit basis vector. Note that the coordinate-wise gradient estimator is commonly taken in the zeroth-order variance reduce algorithms such as in Ji et al. (2019) for minimization problems.

(3) ZO-SREDA-Boost replaces ConcaveMaximizer (line 12 of SREDA) by ZO-ConcaveMaximizer (see Algorithm 5), in which the zeroth-order gradient estimators are recursively updated with small batches $S_{2,x}$ (for update of $x$) and $S_{2,y}$ (for update of $y$) based on the Gaussian estimators given by $G_{\mu_1}(x, y, \nu_{\mathcal{M}_{1,x}}, \xi_{\mathcal{M}_x}) = (1/S_{2,x}) \sum_{i \in [S_{2,x}]} [F(x + \mu_1 \nu_i, y, \xi_i) - F(x, y, \xi_i)] \nu_i/\mu_1$ and $H_{\mu_2}(x, y, \omega_{\mathcal{M}_{2,y}}, \xi_{\mathcal{M}_y}) = (1/S_{2,y}) \sum_{i \in [S_{2,y}]} [F(x, y + \mu_2 \omega_i, \xi_i) - F(x, y, \xi_i)] \omega_i/\mu_2$, where $\nu_i \sim N(0, \mathbf{1}_{d_1})$, $\omega_i \sim N(0, \mathbf{1}_{d_2})$ with $\mathbf{1}_d$ denoting the identity matrices with sizes $d \times d$.

## 5.2 CONVERGENCE ANALYSIS OF ZO-SREDA-BOOST

The following theorem provides the query complexity of ZO-SREDA-Boost for finding a first-order stationary point of $\Phi(\cdot)$ with $\epsilon$ accuracy.

**Theorem 2.** *Apply ZO-SREDA-Boost in Algorithm 4 to solve the online case of the problem eq. (1). Suppose Assumptions 1-4 hold. Let $\zeta = \kappa^{-1}$, $\alpha = \mathcal{O}(\kappa^{-1}\ell^{-1})$, $\beta = \mathcal{O}(\ell^{-1})$, $q = \mathcal{O}(\epsilon^{-1})$, $m = \mathcal{O}(\kappa)$, $S_1 = \mathcal{O}(\sigma^2 \kappa^2 \epsilon^{-2})$, $S_{2,x} = \mathcal{O}(d_1 \kappa \epsilon^{-1})$, $S_{2,y} = \mathcal{O}(d_2 \kappa \epsilon^{-1})$, $\delta = \mathcal{O}((d_1 + d_2)^{0.5} \kappa^{-1} \ell^{-1} \epsilon)$, $\mu_1 = \mathcal{O}(d_1^{-1.5} \kappa^{-2.5} \ell^{-1} \epsilon)$ and $\mu_2 = \mathcal{O}(d_2^{-1.5} \kappa^{-2.5} \ell^{-1} \epsilon)$. Then for $T$ to be at least at the order of $\mathcal{O}(\kappa \epsilon^{-2})$, Algorithm 4 outputs $\hat{x}$ that satisfies $\mathbb{E}[\|\nabla \Phi(\hat{x})\|_2] \leq \epsilon$ with the overall function query complexity $\mathcal{O}((d_1 + d_2) \kappa^3 \epsilon^{-3})$.*

Furthermore, ZO-SREDA-Boost is also applicable to the finite-sum case of the problem eq. (1), by replacing the large batch sample $S_1$ used in line 6 of Algorithm 4 with the full set of samples.

**Corollary 2.** *Apply ZO-SREDA-Boost described above to solve the finite-sum case of the problem eq. (1). Suppose Assumptions 1-4 hold. Under appropriate parameter settings given in Appendix C.6, the function query complexity to attain an $\epsilon$-stationary point is $\mathcal{O}((d_1 + d_2)(\sqrt{n}\kappa^2 \epsilon^{-2} + n) + d_2(\kappa^2 + \kappa n) \log(\kappa))$ for $n \geq \kappa^2$, and $\mathcal{O}((d_1 + d_2)(\kappa^2 + \kappa n)\epsilon^{-2})$ for $n \leq \kappa^2$.*

Theorem 2 and Corollary 2 provide the first convergence analysis and the query complexity for the variance-reduced zeroth-order algorithms for min-max optimization. These two results indicate that the query complexity of ZO-SREDA-Boost matches the optimal dependence on $\epsilon$ of the first-order algorithm SREDA-Boost in Theorem 1 and Corollary 1. The dependence on $d_1$ and $d_2$ typically arises in zeroth-order algorithms due to the estimation of gradients with dimensions $d_1$ and $d_2$. Furthermore, in the online case, ZO-SREDA-Boost outperforms the best known query complexity dependence on $\epsilon$ among the existing zeroth-order algorithms by a factor of $\mathcal{O}(1/\epsilon)$. Including the conditional number $\kappa$ into consideration, SREDA-Boost outperforms the best known query complexity achieved by ZO-SGDMA in the case with $\epsilon \leq \kappa^{-1}$ (see Table 1). Furthermore, Corollary 2 provides the first query complexity for the finite-sum zeroth-order min-max problems.

As a by-product, our analysis of ZO-SREDA-Boost also yields the convergence rate and the query complexity (see Lemma 21) for ZO-iSARAH for the conventional minimization problem, which provides the first complexity result for the zeroth-order recursive variance reduced algorithm SARAH/SPIDER for strongly convex optimization (see Appendix C.4 for detail).

## 6 EXPERIMENTS

Our experiments focus on two types of comparisons. First, we compare SREDA-Boost with SREDA to demonstrate the practical advantage of SREDA-Boost. Second, we compare our proposed zeroth-

order variance reduction algorithm ZO-SREDA-Boost with the other existing zeroth-order stochastic algorithms and demonstrate the superior performance of ZO-SREDA-Boost.

Our experiments solve a distributionally robust optimization problem, which is commonly used for studying min-max optimization Lin et al. (2019); Rafique et al. (2018). We conduct the experiments on three datasets from LIBSVM Chang & Lin (2011). The details of the problem and the datasets are provided in Appendix A.

**Comparison between SREDA-Boost and SREDA:** We set $\epsilon = 0.001$ for both algorithms. For SREDA, we set $\alpha_t = \min\{\epsilon/\|v_t\|_2, 0.005\}$ as specified by the algorithm, and for SREDA-Boost, we set $\alpha_t = 0.005$ as the algorithm allows. It can be seen in Figure 1 that SREDA-Boost enjoys a much faster convergence speed than SREDA due to the allowance of a large stepsize.

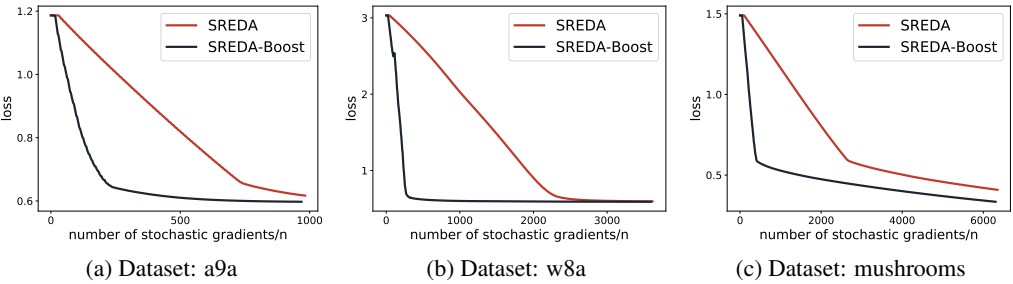

(a) Dataset: a9a        (b) Dataset: w8a        (c) Dataset: mushrooms

Figure 1: Comparison of the convergence rate between SREDA-Boost and SREDA.

**Comparison among zeroth-order Algorithms:** We compare the performance of our proposed ZO-SREDA-Boost with that of two existing stochastic algorithms ZO-SGDA Wang et al. (2020) and ZO-SGDMSA Wang et al. (2020) designed for nonconvex-strongly-concave min-max problems. For ZO-SGDA and ZO-SGDMSA, as suggested by the theorem, we set the mini-batch size $B = Cd_1/\epsilon^2$ and $B = Cd_2/\epsilon^2$ for updating the variables $x$ and $y$, respectively. For ZO-SREDA-Boost, based on our theory, we set the mini-batch size $B = Cd_1/\epsilon$ and $B = Cd_2/\epsilon$ for updating the variables $x$ and $y$, and set $S_1 = n$ for the large batch, where $n$ is the number of data samples in the dataset. We set $C = 0.1$ and $\epsilon = 0.1$ for all algorithms. We further set the stepsize $\eta = 0.01$ for ZO-SREDA-Boost and ZO-SGDMSA. Since ZO-SGDA is a two time-scale algorithm, we set $\eta = 0.01$ as the stepsize for the fast time scale, and $\eta/\kappa^3$ as the stepsize for slow time scale (based on the theory) where $\kappa^3 = 10$. It can be seen in Figure 2 that ZO-SREDA-Boost substantially outperforms the other two algorithms in terms of the function query complexity (i.e., the running time).

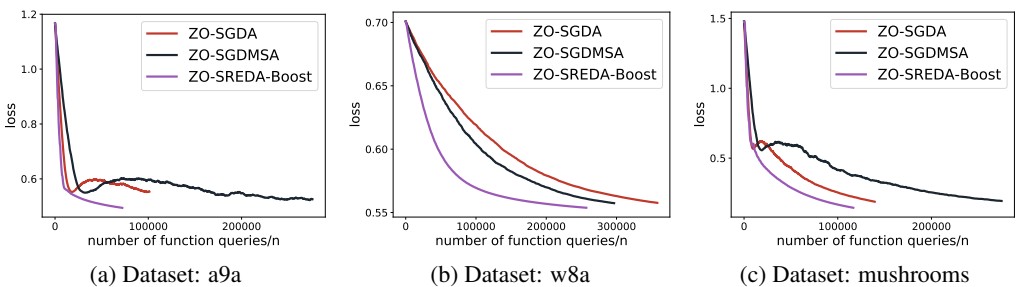

(a) Dataset: a9a        (b) Dataset: w8a        (c) Dataset: mushrooms

Figure 2: Comparison of function query complexity among three algorithms.

# 7 CONCLUSION

In this work, we have proposed enhanced variance reduction algorithms, which we call SREDA-Boost and ZO-SREDA-Boost, for solving nonconvex-strongly-concave min-max problems. In specific, SREDA-Boost requires less initialization effort and allows a large stepsize. Moreover, The complexity of SREDA-Boost and ZO-SREDA-Boost achieves the best complexity dependence on the targeted accuracy among their same classes of algorithms. We have also developed a novel analysis framework to characterize the convergence and computational complexity for the variance reduction algorithms. We expect such a framework will be useful for studying various other stochastic min-max problems such as proximal, momentum, and manifold optimization.

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
