# OpenReview forum: "Enhanced First and Zeroth Order Variance Reduced Algorithms for Min-Max Optimization"
_ICLR.cc/2021/Conference — Reject_

### Official Review · AnonReviewer3 · 2020-10-26
**Enhanced First and Zeroth Order Variance Reduced Algorithms for Min-Max Optimization**

**Rating:** 4
**Confidence:** 5

**Review:**

This paper proposes an enhanced variant of the SREDA algorithm (Lou et al 2020), called SREDA-Boost, that improves SREDA on two aspects: the initial complexity and the step-size. The algorithm achieves the same complexity as the original SREDA scheme.

The main contribution of this paper is perhaps the following aspects:
-- C1: Improving the initial complexity for finding a starting point y0 of the SREDA.
-- C2: Proposing a larger stepsize for SREDA compared to the original one.
-- C3: Injecting a zero-order approximation step for stochastic gradients.
The authors also claimed that their analysis is new and different from SREDA. However, in my opinion, this seems to be minor since their proof also relies on the bounds of gradient errors (like variance) as well as the delta_t quantity. There are of course some technical details and steps, but those are not the major contribution.

In my opinion, the theoretical contribution of this paper is incremental. Indeed, the idea of using SARAH to improve oracle complexity has been widely studied in the literature, including Spider, SpiderBoost, ProxSARAH, etc. Since model (1) is nonconvex-strongly concave, it can be reformulated into (2) as a stochastic optimization problem. Several methods can be used to solve (2). The idea of using multiple loops is also widely used.

The first contribution (C1) is not really new. The authors simply replace the step of computing y0 by iSARAH to reduce the computational cost. Because the problem is strongly convex, this step can be done by several methods, including accelerated variance-reduced schemes to further improve its complexity. Since this step is not essential, many previous methods just simply skip it.

The second contribution (C2) is also minor since the idea of using a large batch-size to obtain large step-size has been used in the literature such as Spider-Boost (Wang et al 2019) or ProxSarah (Pham et al 2020). Of course, this enhancement helps SREDA have better practical performance. However, given the previous work, this contribution is very incremental.
The use of zero-order oracle (C3) is not new as well, since it is another way of approximating the gradients, and it has been widely used in the literature based on Nesterov's idea. Hence, this step seems to be unnecessary if we assume that the underlying function is L-smooth. In most applications, we can directly compute the stochastic gradient components without using finite difference approximation. Unless the authors can provide compelling examples showing that this is an important step, otherwise, it is not really convinced.

In terms of algorithm, SREDA/SREDA-Boost is also a multiple-loop algorithm, with at least three loops, making it challenging to implement in practice and it requires a lot of proper tuning and choosing parameters. There are some recent algorithms that can solve the same problems but with a single loop. The authors may want to compare with them, though these are very recent work (see, e.g. https://arxiv.org/abs/2008.08170 and the references therein).

In addition to the above major comments, the following are some concrete comments:
--- In Table 1, why NA is put in the "Initial Complexity" column of other methods. I believe that some paper may not describe this clearly, but if they use the exact solution (or up to a given accuracy approximation) of the strongly-concave max-problem, then the complexity is the best one since we can use the best algorithm to find it.
-- The relation between the gradient of $\Phi$ and the KKT point of (1) should be clarified.
-- I do not see where v_t is used in Algorithm 2. It seems that the output of Algorithm 2 should include v_t to use in Algorithm 1.
-- It is not clear what is the problem solved in the numerical experiments?

---

### Official Review · AnonReviewer2 · 2020-10-27
**Some interesting new results are made, but the presentation needs to be improved**

**Rating:** 6
**Confidence:** 5

**Review:**

This paper studies the nonconvex strongly-concave min-max optimization problem. It improves the analysis of an existing method SREDA to make it allow larger step size and less initialization computation. Besides, it extends the algorithm to the case where the objective function is non-differentiable. The authors claimed it is the first zeroth-order variance-reduced method for the min-max problem. Experiments are conducted to demonstrate the improved algorithm is better than existing methods.

The results of this paper are interesting. However, I don't think the initialization complexities really matter, since they are always dominated by the complexity of the later optimization process. Hence, improving initialization complexity does not make any change to the overall complexity of the whole algorithm. I think the key to making SREDA-Boost performs better than SREDA is the larger step size.

The presentation of this paper, especially for the algorithms, need to be improved. In line 3 of Algorithm 1, function iSARAH is called. However, the meaning of its two parameters is not defined. It is unclear which variables these two parameters correspond to in iSARAH (Algorithm 3 in appendix). Besides, the \tilde{v}_{t-1,\tilde{m}_{t-1}} and \tilde{u}_{t-1,\tilde{m}_{t-1}} in line 8 are not defined there. Though these two variables are introduced in Algorithm 2, algorithms need to be self-contained. Furthermore, when $t=0$, line 8 seems wrong.

---

### Official Review · AnonReviewer4 · 2020-10-28
**Marginal contribution**

**Rating:** 5
**Confidence:** 3

**Review:**

The paper proposes a SREDA-Boost, which builds upon SREDA for nonconvex-strongly-concave minimax problem. The SREDA-Boost algorithm is less restrictive to initialization and has an accuracy-independent and larger step size. Thus it can run substantially faster than SREDA. The main contribution to the first-order optimization story is a new analytical framework that builds upon the previous analysis in SREDA and overcomes the dependence of highly accurate initialization via bounding the tracking error and gradient estimation error separately. It also proposes a zeroth-order variance reduction algorithm for the same optimization problem, which has the largest possible step size so far and also improves the complexity of the state-of-the-art in some cases. Various experiments have validated the superiority. The theoretical analysis and empirical results look good to me.


My main concern is the significance of the contribution. It seems that SREDA-Boost is basically the same as SREDA except for stepsize and initialization complexity, which makes the contribution marginal. Though the analysis is definitely new. For the zero-th order optimization problem, it seems that the previous algorithm ZO-SGDMSA does not require good initialization and has the same step size. The complexity of it is also comparable to ZO-SREDA-Boost when kappa is approximately 1/eps. It would be better if the authors could discuss a bit more the initialization requirement of both algorithms and the query complexity, along with the novelty in both first order and zero-th order story.

---

### Official Review · AnonReviewer1 · 2020-10-30
**The paper is technical, further explanation would be helpful for better understanding of the result**

**Rating:** 6
**Confidence:** 3

**Review:**

The paper proposes a variant SREDA-Boost of the variance reduction method SEDRA for solving nonconvex-strongly-concave min-max problem. The first contribution of the paper is to relax the conditions on the initialization  of SEDRA and moreover enable larger stepsizes ($\epsilon$-independent stepsizes). As SEDRA is already optimal, such modification does not improve the theoretical convergence rate, but it is beneficial from the practical perspective. The second contribution is to adapt the method to zero order oracle, achieving the state-of-the-art convergence rate.

The result are presented in a clear and technical manner. My major concern is the lack of high level explanation on why a larger stepsize can be applied. I understand that the paper introduces a novel way for the complexity analysis, however the proof is very long and not easy to check. Hence, it would be helpful to explain in words the key aspects that allows an $\epsilon$-independent stepsize (in a non-technical manner).

Another question I would like to ask is whether the Boost version is necessary to deduce the zeroth order variant. In other words, what would be the complexity of ZO-SREDA. I would expect that it shares the same complexity as  ZO-SREDA-BOOST, even though the stepsize are smaller. Please clarify this point.

The experiments are rather limited but it is fine for a theoretical paper.  I would expect including more comparisons on competitive methods in Figure 1, as in the current version the only baseline is ZO-SREDA.

---

### Decision · Program_Chairs · 2021-01-07
**Final Decision**

**Decision:**

Reject

**Comment:**

The paper introduces a new variant (SREDA-Boost) of a variance-reduced method SEDRA for nonconvex-strongly-concave min-max optimization. Given that SEDRA is already optimal in the worst case, the proposed modification is intended to improve practical performance of the method, by relaxing conditions needed at initialization and allowing larger step sizes. While the reviewers appreciated the main ideas of the paper, they shared concerns about the significance of the paper's technical contributions, which were ultimately not addressed by the authors in the rebuttal phase.